# Influence of Cr, Mn, Co and Ni Addition on Crystallization Behavior of Al_13_Fe_4_ Phase in Al-5Fe Alloys Based on ThermoDynamic Calculations

**DOI:** 10.3390/ma14040768

**Published:** 2021-02-06

**Authors:** Na Pang, Zhiming Shi, Cunquan Wang, Ninyu Li, Yaming Lin

**Affiliations:** School of Materials Science and Engineering, Inner Mongolia University of Technology, Hohhot 010051, China; wcq94810@163.com (C.W.); liningyu518@163.com (N.L.); liny00@163.com (Y.L.)

**Keywords:** Al_13_Fe_4_ phase, 3d transition elements, Gibbs free energy, formation enthalpy, crystallization behavior

## Abstract

Alloying is an effective method to refine coarse grains of an Al_13_Fe_4_ phase and strengthen Al-Fe alloys. However, the grain refinement mechanism remains unclear in terms of the thermodynamics. Herein, the influence of *M*-element, i.e., Cr, Mn, Co and Ni, addition on the activity of Al and Fe atoms, Gibbs free energy of the Al_13_Fe_4_ nucleus in Al-Fe melt and the formation enthalpy of an Al_13_Fe_4_ phase in Al-Fe alloys is systematically investigated using the extended Miedema model, Wilson equation, and first-principle calculations, respectively. The results reveal that the addition of different M elements increases the activity of Fe atoms and reduces the Gibbs free energy of the Al_13_Fe_4_ nucleus in Al-Fe melt, where the incorporation of Ni renders the most obvious effect, followed by Mn, Co, and Cr. Additionally, the formation enthalpy decreases in the following order: Al_78_(Fe_23_Cr) > Al_78_(Fe_23_Mn) > Al_13_Fe_4_ > Al_78_(Fe_23_Ni) > Al_78_(Fe_23_Co), where the formation enthalpy of Al_78_(Fe_23_Ni) is close to Al_78_(Fe_23_Co). Moreover, the presence of Ni promotes the nucleation of the Al_13_Fe_4_ phase in Al-Fe alloys, which reveals the mechanism of grain refinement from a thermodynamics viewpoint.

## 1. Introduction

Al-Fe alloys are promising candidates for high-temperature applications due to the presence of a thermodynamically stable Al_13_Fe_4_ phase in the aluminum (Al) matrix [1,2]. However, the mechanical properties of Al-Fe alloys are compromised due to the coarse flake-like, needles and laths of the Al_13_Fe_4_ phase [3,4,5]. To date, different approaches have been adopted to enhance the mechanical properties of Al-Fe alloys by reducing the grain size and improving the morphology and distribution of the Al_13_Fe_4_ phases.

Transition metals are widely used as dopants due to their low solid solubility and diffusion rate in Al-based alloys [6], forming coarsening-resistant intermetallics, such as ternary Al-Fe-X alloys (X = Zr, Mo, Si, Ni and Cr). These ternary Al-Fe-X alloys render high strength at temperatures up to 400 °C [7,8]. For instance, the addition of Ni refines the Al_13_Fe_4_ phase and promotes the formation of orthorhombic Al_3_Ni and monoclinic Al_9_FeNi phases [8,9,10], improving the high-temperature mechanical properties and decreasing the thermal expansion coefficient (TEC) of the Al-Fe alloys [11,12]. Additionally, instead of forming an AlCr binary compound, Cr prefers to dissolve in the Al_13_Fe_4_ phase and improve the morphology, whereas the presence of Mn stabilizes the metastable Al_6_Fe phase and forms an Al_6_(Fe,Mn) solid-solution. With the increase in Cr content, the morphology of the Al_13_Fe_4_ phase changes from a needle-like structure to poly-angled and sheet-like structures in Al-5Fe alloys [13]. Furthermore, the addition of Mn increases the yield strength and ultimate tensile strength of Al-Fe alloys, however, the elongation is compromised due to the formation of a large volume fraction of Fe- and Mn-bearing intermetallics, in addition to crack propagation [14]. It has been reported that the addition of 0.2 wt% Co completely dissolved in the Al_13_Fe_4_ phase and significantly improved the morphology of the Al_13_Fe_4_ phase in Al-5Fe alloys [15].

These experimental studies demonstrate the prominent influence of alloying on the grain refinement of an Al_13_Fe_4_ phase and the mechanical properties of Al-Fe alloys. One should note that the reinforcement effect of the Al_13_Fe_4_ phase in Al-Fe alloys depends on nucleation and coarsening, which are influenced by the changes in the formation enthalpy and Gibbs free energy. The alloying elements influence the Gibbs free energy by altering the component activity in the Al-Fe melt and affect the formation enthalpy by forming a solid-solution. The decrease in Gibbs free energy and formation enthalpy facilitates phase transformation, sub-cooling degree and nucleation rate, promoting the reinforcement of the Al_13_Fe_4_ phase. However, the component activity, Gibbs free energy and formation enthalpy of multicomponent melts are rarely reported due to the complex and extended lab-scale experiments involved.

Therefore, it is of utmost significance to develop a theoretical approach to screen alloying elements for the refinement of the Al_13_Fe_4_ phase. Herein, theoretical calculations are used to investigate the influence of M addition, where M refers to Cr, Mn, Co, or Ni, on the Gibbs free energy and formation enthalpy of the Al_13_Fe_4_ phase. First, the effect of M incorporation on the activity of Al and Fe atoms and the Gibbs free energy of the Al_13_Fe_4_ phase in an Al-Fe melt is estimated by using the extended Miedema model and Wilson equation, respectively. Second, the influence of M addition on the formation enthalpy of the Al_13_Fe_4_ phase is assessed using the first-principle calculations. Third, the microstructural evolution of Al-Fe alloys with the addition of Ni and Cr is investigated based on theoretical predictions. The current study aimed to reveal the mechanism of grain refinement of the Al_13_Fe_4_ phase due to the addition of 3D transition elements from a thermodynamics viewpoint.

## 2. Calculations and Experimental Procedures

### 2.1. Change in Gibbs Free Energy of Al-Fe Melt

A hypereutectic Al-5wt%Fe (Al-2.47at%Fe) alloy and corresponding melt (at 850 °C) were selected as objects for thermodynamics calculations, where the melt temperature was slightly greater than the crystallization temperature. The formation of a coarse Al_13_Fe_4_ phase can be given as:13[Al] + 4[Fe] = Al_13_Fe_4_(s),(1)

The change in Gibbs free energy during the formation of primary Al_13_Fe_4_ phase can be expressed as:(2)$$\Delta G_{Al_{13}Fe_{4}}^{1}=\Delta G_{Al_{13}Fe_{4}}^{0}-13RT\ln a_{Al}-4RT\ln a_{Fe},$$
where $\Delta G_{Al_{13}Fe_{4}}^{0}$ refers to the standard Gibbs energy of formation, *a_i_* represents the component activity, *R* corresponds to a gas constant and *T* denotes the absolute temperature of the Al-Fe melt. The activity of the Al_13_Fe_4_ solid in the melt can be considered as 1.

The incorporation of the M element forms a ternary Al-Fe-M system, where the activity of both Al and Fe atoms is also altered. Therefore, Equation (2) can be rearranged as:(3)$$\begin{array}{cc} \Delta G_{Al_{13}Fe_{4}}^{2} & =\Delta G_{Al_{13}Fe_{4}}^{0}-13RT\ln\left(a_{Al}+\Delta a_{Al}\right)-4RT\ln\left(a_{Fe}+\Delta a_{Fe}\right)\\ & =\Delta G_{Al_{13}Fe_{4}}^{0}-13RT\ln\left(x_{Al}+\Delta x_{Al}\right)\left(r_{Al}+\Delta r_{Al}\right)-4RT\ln\left(x_{Fe}+\Delta x_{Fe}\right)(r_{Fe}+\Delta r_{Fe}),\\ & =\Delta G_{Al_{13}Fe_{4}}^{0}-13RT\ln\left(1+P_{Al}\right)\left(1+Q_{Al}\right)-4RT\ln\left(1+P_{Fe}\right)\left(1+Q_{Fe}\right) \end{array}$$
where *a_i_* = *x_i_ ∙ r_i_, x_i_* represents the atomic fraction of component *i* in the melt and *γ_i_* refers to the activity coefficient. Δ*x_i_* and Δ*r_i_* represent the change in component concentration and activity coefficient, respectively. Additionally, *P_i_* and *Q_i_* can be given as:(4)$$P_{i}=\frac{\Delta x_{i}}{x_{i}},\;Q_{i}=\frac{\Delta r_{i}}{r_{i}},$$
Then, using the prediction model, the activity coefficient in a multicomponent system can be theoretically calculated. Combining Equations (2) and (3), the Gibbs free energy change, ∆*G*, can be calculated with alloying elements additions. Therefore, the influence of alloying element additions on the chemical reaction can be determined.

The aforementioned theoretical models for calculating chemical activity are usually suitable for binary alloys. Fan et al. [16,17,18] utilized Wilson’s equation [19] to propose a novel method for calculating the activity of components in a multicomponent melt. According to the extended Miedema model [20] and Wilson equation [19], which only relies on the physical parameters of alloying elements, the reliability of this method was well verified in Al-Mg-X (X=Si, Mn, Cu, Zn), Al-Ti-B, Al-Ti-B-X (X=Mg, Si, Cu, Zr, V, Fe, Ni, La) systems [16,17,21].

According to the thermodynamic model, introduced by Fan et al., the activity coefficient (*γ_i_*) in *i*, *j*, and *k* ternary systems can be given as:(5)$$\begin{array}{l} \ln\gamma_{i}=1-\ln(1-x_{j}A_{j/i}-x_{k}A_{k/i})-\frac{x_{i}}{1-x_{j}A_{j/i}-x_{k}A_{k/i}}\\ -\frac{x_{j}(1-A_{i/j})}{1-x_{i}A_{i/j}-x_{k}A_{k/j}}-\frac{x_{k}(1-A_{i/k})}{1-x_{i}A_{i/k}-x_{j}A_{j/k}} \end{array},$$
where *A_i/j_* and *A_j/i_* are adjustable parameters. The pair of parameters, i.e., *A_i/j_* and *A_j/i_*, can be calculated by $\ln\gamma_{i}^{x_{i}\to 0}$ and $\ln\gamma_{j}^{x_{j}\to 0}$; which are based on the binary infinitely dilute activity coefficients, as given below:(6)$$\ln\gamma_{i}^{x_{i}\to 0}=-\ln(1-A_{j/i})+A_{i/j},$$
(7)$$\ln\gamma_{j}^{x_{j}\to 0}=-\ln(1-A_{i/j})+A_{j/i},$$

According to the Miedema model and thermodynamics model, the activity coefficient of component *i* in an infinite solution *j* can be given as:(8)lnγixi→0=αijfij[1+ui(φi−φj)]RTVj23,

Here, *α_ij_* and *f_ij_* are defined as:(9)$$\alpha_{ij}=1-0.1T\left(\frac{1}{T_{m_{i}}}+\frac{1}{T_{m_{j}}}\right)$$
(10)fij=2pVi23Vj23{q/p[(nws13)i−(nws13)j]2−(φi−φj)2−b(r/p)}(nws13)i−1+(nws13)j−1

The advantage of this method is that, independent of the experimental data, it is applicable to multiple liquid alloys and capable of predicting thermodynamic data according to the physical parameters of these elements. In Equations (8)–(10), $T_{m_{i}}$ and $T_{m_{j}}$ represent the melting temperature of component *i* and *j*, respectively; *φ* refers to electronegativity in volts; *n_ws_* denotes the electron density at the boundary of the Wigner–Seitz cell in density units (d.u., about 6 × 10^22^ electrons/cm^3^); *V* refers to the molar volume in cm^3^/mol, and *u* is a constant. For all alloys, q/p is equal to 9.4 V^2^/(d.u.)^2/3^. The values of *p* for alloys of two transition metals, two non-transition metals, and a transition metal with a non-transition metal are 14.2, 10.7 and 12.35, respectively. The term *b* for the solid, liquid alloy with a transition metal and a non-transition metal, and other alloys is equal to 1.0, 0.73 and 0, respectively.

### 2.2. Formation Enthalpy of Al_13_Fe_4_ Phase Based on First-Principle Calculations

#### 2.2.1. Crystal Structure of Al_13_Fe_4_

The preliminary X-ray diffraction results suggest that the Al_13_Fe_4_ compound possesses a monoclinic structure with a complex bottom center [22,23,24], which belongs to the space group C2/m. The unit cell of Al_13_Fe_4_ is composed of twenty crystallographically different atomic species shown in Table 1, containing 15 different Al atoms and 5 different Fe atoms (*a* = 15.489 Å, *b* = 8.0831 Å, *c* = 12.476 Å, *β* = 107.72°). In total, 102 atoms (78 Al and 24 Fe) are shown in Figure 1. The coordination number of Fe-1, Fe-2 and Fe-5 is 11, 10 and 9, respectively, whereas the coordination number of Fe-3 and Fe-4 is 10 with Al and 1 with Fe [22].

#### 2.2.2. Computational Details

The calculations were performed via the Materials Studio program using the Cambridge sequential total energy package (CASTEP) based on the density-functional theory (DFT) [25], where a combination of generalized gradient approximation (GGA) and Perdew–Burke–Ernzerhof (PBE) was employed to the electronic exchange-correlation energy [26], and the ultra-soft pseudopotential was employed to describe ionic interactions [27].

Based on the convergence test, the *k*-point of Al_13_Fe_4_ and Al_78_(Fe_23_M) compound was set as 2 × 3 × 2, which was generated by the Monkhorst-Pack scheme [28]. The substituted compounds are represented by the actual number of atoms in a cell. The cut-off energy was set at 400 eV. The self-consistent field (SCF) method was employed to calculate the total energy based on the Pulay density mixing method. The geometric optimization was performed using the Broyden–Flecher–Goldfarb–Shanno (BFGS) method to obtain the most stable structure [29]. Herein, the total energy of 10^−5^ eV/atom and the maximum force tolerance of 0.03 eV/Å were set for optimization.

Moreover, the formation enthalpy of Al_78_(Fe_23_M) under different spin directions and spin states was calculated. The results revealed that the consideration of magnetism rendered a negligible influence and the effect of four elements on the Al_78_(Fe_23_M) phase remained unchanged. Therefore, the spin polarization was not considered because it is unnecessary for such Al-rich complex intermetallic compounds [30]. Hence, the magnetism was not considered in subsequent calculations to optimize the utilization of available computing resources.

The formation enthalpy (∆*H*) of Al_78_(Fe_23_M) (M = Cr, Mn, Co or Ni) alloys at 0 K can be defined as:(11)$$\Delta H=\frac{1}{102}(E_{tot}^{Al_{78}(Fe_{23}M)}-78E_{solid}^{Al}-23E_{solid}^{Fe}-E_{solid}^{M}),$$
where $E_{tot}^{Al_{78}(Fe_{23}M)}$, $E_{solid}^{Al}$, $E_{solid}^{Fe}$ and $E_{solid}^{M}$ represent the total energy of the Al_78_(Fe_23_M), Al, Fe and M (M = Cr, Mn, Co, Ni), respectively.

### 2.3. Experimental Details

To confirm the effectiveness of M addition on grain refinement of the Al_13_Fe_4_ phase, an Al-5wt%Fe alloy was prepared by melting pure aluminum (99.7 wt. %) and master alloys of Al-20wt%Fe, Al-10wt%Ni and Al-5wt%Cr to prepare Al-5wt%Fe-1wt%M (*M* = Ni and Cr), respectively. For each composition, the aluminum ingot was melted in an electrical resistance furnace and the Al-20wt%Fe ingot was added to form the Al-5wt%Fe melt, at 880 °C for 40 min. Then, different amounts of Al-10wt%Ni and Al-5wt%Cr master alloys were added into the molten alloy and the melts were stirred for 15 min using a graphite stirrer. The nitrogen was utilized to eliminate gases and avoid oxidation of the alloy melt. When the temperature decreased to 850 °C, the melts were poured into bars (Φ35 × 25 mm) for microstructural observations.

The as-cast ingots were cooled down to room temperature in the mold and sectioned at the tip in the thermocouple sheath. The metallographic samples were rubbed with sandpaper and then polished with a polishing machine. Five micrograph views were chosen for length measurement of Al_13_Fe_4_. Each view obtained around 50 data points for average calculation. For each component, three samples were prepared and microstructural observations were carried out from the same position at the thermocouple tip. A scanning electron microscope (SEM, S3400-N, Hitachi, Tokyo, Japan) at an accelerating voltage of 20 kV under high vacuum, equipped with a 7021-H/Horiba energy dispersive spectrometer (EDS, Kyodo, Japan), was adopted to observe the morphology and analyze the elemental distribution. A D/MAX-2500/PC/PIGAKV X-ray diffractometer (XRD, Rigaku corporation, Tokyo, Japan), equipped with a Cu kα target, was employed to analyze the phase composition. XRD patterns were recorded in the 2θ range of 20° to 70° at the scan speed of 3 °/min.

## 3. Results and Discussion

### 3.1. Gibbs Free Energy of Al_13_Fe_4_ Nucleus in Al-Fe Melt

The component activity is an important factor in determining thermodynamic stability of the reaction between the Al_13_Fe_4_ phase and the Al matrix in liquid alloys. Thus, it is vital to understand this mechanism theoretically and thereby control it favorably, especially in a multiple-component system. When the Al-Fe melt includes alloying components, based on Equation (5), the investigation on the influence of alloying element additions on the chemical stability of Equation (1) can be divided into two aspects: (i) the variation *P_i_* of component concentration in the composite melt, and (ii) the impact factor *Q_i_* on activity coefficients of Al and Fe. Al-Fe-M alloy composites, such as alloy melt, at high temperatures can be described as an Al-Fe-M ternary liquid alloy system when the chemical reaction in Equation (1) takes place. The values of the parameters required are presented in Table 2 [31], and according to Equations (6)–(9), the values of Wilson’s parameters were obtained in binary M-Al, M-Fe, and Al-Fe systems at 850 °C (Table 3).

The effect of M addition on the activity coefficient of Al and Fe atoms in the Al-2.47 at%Fe (Al-5wt%Fe) alloy at 850 °C, at the beginning stage of the chemical reaction in Equation (1), is shown in Figure 2. The results indicate that different alloying additions can result in different variations due to the distinct physical characteristics of M elements. With the M content increasing, the activity coefficient of Al decreases to different extents. On the contrary, the activity coefficient of Fe increase to different extents. In addition, Ni addition has a remarkable influence on the activity of Al and Fe, whereas Cr addition has little effect on the activity of Al and Fe. According to the values of activity coefficient, the activity of Al and Fe were obtained and are shown in Figure 3. The influence of M elements on the activity of Al and Fe is consistent with that of the activity coefficient of Al and Fe.

The nature of alloying elements significantly influenced the variation of Gibbs free energy, originating from the physical characteristics of the alloying elements (Figure 4). It can be seen that the Gibbs free energy of the chemical reaction decreases to different extents with different alloying additions. Moreover, with M element addition, the Gibbs free energy can be decreased and this is associated with the increased formation of Al_13_Fe_4_. The addition of Ni can significantly decrease the free energy and visibly promote the reaction. The Gibbs free energy of the reaction in Equation (1) provides a direct driving force for the formation of Al_13_Fe_4_ phase which is the only product in the reaction. The results reveal that the incorporation of alloying elements reduced the Gibbs free energy and, in turn, increased the driving force for phase transition.

It is worth noting, that the precipitation of the Al_13_Fe_4_ phase from the supersaturated solution of Al-2.47at% Fe melt represents a transition from the metastable phase to a stable phase, which is realized by the difference in Gibbs free energy. This difference in Gibbs free energy determines the speed of nucleation and growth. So, the change in Gibbs free energy influences the nucleation and growth of the Al_13_Fe_4_ phase. On the one hand, according to the chemical reaction kinetics, the primary homogeneous nucleation rate can be expressed as [37]:(12)I=102⋅nkBTh⋅exp(−ΔGc/kBT)=102⋅nkBTh⋅exp(−16πσ3V23kBTΔG2),
where *n* represents the number of atoms in a liquid mass, *k_B_* refers to the Boltzmann constant, σ corresponds to the interfacial energy per square decimeter between liquid and solid crystals, *V* denotes the volume of crystals and Δ*G_c_* refers to the free energy for critical nucleus formation. It indicates that the nucleation rate is exponentially related to the Gibbs free energy. It has no effect on the volume (V) and Gibbs free energy for critical nucleus formation (Δ*G_c_*) when 3D transition atoms are not involved in crystallization. Under the same number of 3D transition atoms (*n*) in the melt, the interfacial energy per square decimeter can be considered equal, due to slight differences. So, Equation (12) shows that the change in Gibbs free energy only determines the primary homogeneous nucleation rate.

In the case of Al-2.47at%Fe melt, due to the low solubility of Fe in aluminum [38], the generalized equation for the growth of the crystal nucleus, according to the linear growth rate, can be given as:(13)υ=A⋅ΔG⋅exp(b/T),
where *A* and *b* are system constants [38]. Overall, the nucleation rate is more sensitive to Gibbs free energy than the growth rate. The increase in the absolute value of Gibbs free energy (∆*G* < 0) is conducive to grain refinement. On the other hand, it exhibits a relationship of ∆*G* = ∆*H_m_* − *T_m_*∆*S * under solid–liquid equilibrium conditions. The phase transition begins at below the equilibrium temperature, exhibiting the following relationship [38]:(14)$$\Delta G=\Delta H-\left(T_{m}-\Delta T\right)\Delta S\approx \Delta H_{m}-T_{m}\Delta S+\Delta T\Delta S\approx \Delta T\Delta S,$$
where ∆*H_m_* and *T_m_* represent the latent heat of phase change and crystallization temperature, respectively, and ∆*T* refers to the subcooled degree. Under these conditions, the change in entropy (∆*S*) can be regarded as a constant. Equation (14) indicates that the Gibbs free energy is proportional to the subcooled degree and the increase in the absolute value of the Gibbs free energy is conducive to an increase in the subcooled degree, which can promote the refinement of the Al_13_Fe_4_ phase. Hence, it can be predicted that the incorporation of the M element is beneficial for promoting nucleation and refinement of the Al_13_Fe_4_ phase, where Ni renders the most prominent influence, followed by Mn, Co and Cr.

### 3.2. Formation Enthalpy of Al_13_Fe_4_ Phase

In addition to the Gibbs free energy of the Al_13_Fe_4_ phase, the incorporation of transition metal elements affected the formation enthalpy of the Al_13_Fe_4_ phase by forming substitutional solid-solutions. The effect of M addition on the formation enthalpy of the Al_13_Fe_4_ phase was assessed using the first-principle calculations. First, the Al_13_Fe_4_ crystal was optimized to ensure the reliability of the calculations. The optimized lattice constants of the Al_13_Fe_4_ crystal were *a* = 15.366 Å, *b* = 8.012 Å, *c* = 12.393 Å and *β* = 107.67°, which are close to the previously reported values (*a* = 15.487 Å, *b* = 8.0831 Å, *c* = 12.476 Å and *β* = 107.72°) which was obtained from powder XRD [22].

To investigate the occupying tendency of the M element in the Al_13_Fe_4_ phase, the formation enthalpy of Al_78_(Fe_23_M) (M = Cr, Mn, Co and Ni) was calculated and is shown in Figure 5. As Cr, Mn, Co, and Ni belong to the same row of the periodic table and lie next to each other, these elements exhibit a small difference in atomic radii. Hence, M elements easily substituted Fe-sites in the Al_13_Fe_4_ phase. It has been reported that these four elements are more likely to occupy the Fe sites [39,40,41,42]. If M substitutes one Fe atom in the Al_13_Fe_4_ unit cell, no phase transition occurs and the proportion of substituted atoms remains at ~0.98 at%. The lower value of formation enthalpy corresponds to the more stable crystal structure [43]. The negative values of formation enthalpy of Al_78_(Fe_23_M) (*M* =Cr, Mn, Co and Ni) indicate that the substituted compounds are thermodynamically stable. The formation enthalpy decreased in the given order: Al_78_(Fe_23_Cr) > Al_78_(Fe_23_Mn) > Al_13_Fe_4_ > Al_78_(Fe_23_Ni) > Al_78_(Fe_23_Co), where the formation enthalpy of Al_78_(Fe_23_Ni) is comparable to the Al_78_(Fe_23_Co) phase.

Additionally, the Cr and Mn preferred to occupy the Fe-5 site, whereas Co and Ni preferred the Fe-1 sites. Compared with Fe, the 3D orbitals of Co and Ni contain fewer electrons, which favors the occupation of Fe-1 site with a large coordination number, whereas the Cr and Mn occupy the Fe-5 position with a smaller coordination number. It can be inferred that the number of electrons in the 3D orbitals of transition metals critically influences the substitution position. It has been reported that the contribution of Fe in different positions at the Fermi level is different, which is related to the difference in the coordination number [44,45,46]. The inter-atomic bonding process is exothermic [47]. Therefore, the larger coordination number corresponds to the higher bonding degree and better stability. The addition of Co and Ni decreased the formation enthalpy by occupying the Fe-1 position with a large coordination number. Additionally, the formation of several bonds is more conducive to increasing compound stability.

On the other hand, the addition of Cr and Mn increased the formation enthalpy of the compound and decreased its stability. The replacement of Fe-5 positions with a smaller coordination number and fewer bonds is more conducive to slowing down the increase in formation enthalpy. Based on the formation enthalpy results, the interatomic forces of the compounds, which are formed after the substitution of Cr and Mn, are reduced, where the bond length of Al-M is found to be higher than that of Al-Fe. Additionally, the stability of Al_78_(Fe_23_Cr) and Al_78_(Fe_23_Mn) compounds is compromised, and both Cr and Mn are more likely to occupy the Fe-5 position. On the other hand, the interatomic forces of Al_78_(Fe_23_Ni) and Al_78_(Fe_23_Co) compounds, which are formed after Ni and Co substitution, are enhanced and the Al-M bond length became shorter than the Al-Fe. Moreover, the thermal stability of Al_78_(Fe_23_Ni) and Al_78_(Fe_23_Co) compounds is increased, and both Ni and Co preferred to occupy the Fe-I position with a large influence on the overall energy.

The lattice constants of the Al_13_Fe_4_ and Al_78_(Fe_23_M) phases, with M = Cr, Mn, Co and Ni, which possesses the highest negative value of formation enthalpy, are shown in Table 4. It can be readily observed that the introduction of M elements induced lattice distortions and increased the unit cell volume. The increase in unit cell volume remained consistent with the atomic radii of M elements. It is worth noting that, in addition to the Al_78_(Fe_23_Co) phase, the formation enthalpy and volume change trends of Al_78_(Fe_23_M) phases (*M* = Cr, Mn, Ni) are consistent with the change in atomic radii of M elements. If only the effect of atomic radius on volume is considered, the volume of Al_78_(Fe_23_Co) should be larger than the Al_78_(Fe_23_Ni) phase, which is contrary to our current results. Hence, compared with the atomic radii, the formation enthalpy plays a more prominent role in defining the volume of a unit cell. It has been reported that the monoclinic phases of Al_13_Co_4_ and Al_13_Fe_4_ possess similar crystallographic structures and may form a continuous solid-solution of Al_13_(Fe,Co)_4_ [28,48], which is not observed in the case of other transition metals. The experimental and theoretical studies revealed that the formation enthalpy of Al_13_Co_4_ is more negative than the Al_13_Fe_4_ phase [49]. This can be the reason for the lower formation enthalpy and a smaller volume of Al_78_(Fe_23_Co) phase than the Al_78_(Fe_23_M) phase (M = Cr, Mn and Ni). Hence, the incorporation of 3D transition elements changed the lattice constants of the Al_13_Fe_4_ phase in different degrees, but this change is small enough to cause a change in the symmetry of lattice structure.

According to the thermodynamic driving force for phase transitions, the Gibbs free energy under isobaric conditions can be expressed as: G(t) = H(T) − *T*S(T),(15)

Herein, solidification precipitation of the Al_13_Fe_4_ phase from the Al-Fe alloy liquid phase requires a driving force of Δ*G* < 0 (Δ*G* = Δ*H* − *T*Δ*S*). When M replaces one Fe atom under the same temperature and concentration, the slight increase in ΔS can be considered as a constant in the Al_78_(Fe_23_M) phase. It can be predicted that the decrease in formation enthalpy increases the driving force for phase transition after forming Al_78_(Fe_23_Ni) and Al_78_(Fe_23_Co) phases, which is more conducive to promoting nucleation and refinement of the Al_13_Fe_4_ phase. Additionally, the compromised stability promotes the formation of other metastable phases, such as Al_6_Fe, in the Al-Fe alloy after the substitution of Cr and Mn. One should note that the formation of the metastable Al_x_Fe phase in Al-Fe alloys due to the addition of Cr and Mn has been reported [13,41].

### 3.3. Influence of M Addition on Microstructural Evolution

Furthermore, we have selected Al-5wt%Fe-1wt%Cr and Al-5wt%Fe-1wt%Ni alloys, with the minimum and maximum change in Gibbs free energy of Al_13_Fe_4_ phase, and studied the microstructural evolution due to the addition of M element (Figure 6). The length distribution chart of Al_13_Fe_4_ is shown in Figure 7. The length of the primary Al_13_Fe_4_ phase ranges from 10 to 200 μm, which includes some very large coarse grains. The value of the average length is 43.10 μm. The addition of Ni greatly refined the grains of primary Al_13_Fe_4_ phase (Figure 6b), and the average length of primary Al_13_Fe_4_ phase ranges from 5 to 70 μm. The value of the average length is 27.75 μm, whereas the addition of Cr only slightly refined the grains of primary Al_13_Fe_4_ phase (Figure 6c). The value of the average length is 39.25 μm. This is consistent with our previous analysis on the effect of alloying elements, showing that Ni renders the most prominent refinement effect. The mechanism of grain refinement from a thermodynamics viewpoint clearly revealed the experimental phenomena.

Figure 8 presents XRD patterns of the as-cast Ni- and Cr-added alloys. The Al-5wt%Fe-1wt%Ni alloy consisted of α-Al, Al_13_Fe_4_, Al_9_FeNi, and Al_3_Ni phases in the as-cast state (Figure 8a), whereas Al-5wt%Fe-1wt%Cr alloy consisted of α-Al, Al_13_Fe_4_, and Al_13_Cr_2_ phases in the as-cast state (Figure 8b). The added M element partially formed the secondary compound with Al and promoted heterogeneous nucleation of the α-Al phase. The remaining amount of M element was dissolved in the Al_13_Fe_4_ phase, which leads to grain refinement and microstructural changes. The EDS results confirm the presence of Fe, Cr and Ni atoms in α-Al grains (Point-A and Point-D), as shown in Figure 9 and Figure 10. One should note that the solid-solubility of M elements in α-Al is extremely small, whereas the solid-solubility of Cr and Ni in the primary Al_13_Fe_4_ phase is relatively large (Point-B and Point-E). As the amount of eutectic Al_13_Fe_4_ phase is relatively small and the formation temperature is low, a small amount of Cr element was dissolved in the Al_13_Fe_4_ phase (Point-C). Moreover, the Al_9_FeNi phase was distributed at grain boundaries in the form of layers (Point-F), however, its influence on the Al_13_Fe_4_ phase will be studied later.

## 4. Conclusions

In summary, the current study aimed to reveal the mechanism of grain refinement of an Al_13_Fe_4_ phase due to the addition of 3D transition elements, from a thermodynamics viewpoint. The main conclusions of the current study can be summarized as:

(1) M addition increased the activity and effective concentration of Fe, whereas it reduced the activity of Al in Al-Fe alloy melt. Additionally, the incorporation of M elements reduced the Gibbs free energy and increased the driving force for phase transition, promoting the nucleation and refinement of the Al_13_Fe_4_ phase. Moreover, Ni addition rendered the most prominent influence, followed by Mn, Co, and Cr.

(2) The formation enthalpy decreased in the following order: Al_78_(Fe_23_Cr) > Al_78_(Fe_23_Mn) > Al_13_Fe_4_ > Al_78_(Fe_23_Ni) > Al_78_(Fe_23_Co), where the formation enthalpy of Al_78_(Fe_23_Ni) was comparable to the Al_78_(Fe_23_Co). Additionally, Cr and Mn preferred to occupy the Fe-5 sites, whereas Co and Ni preferred to occupy the Fe-1 sites. Overall, the decrease in formation enthalpy increased the driving force of the phase transition after forming Al_78_(Fe_23_Ni) and Al_78_(Fe_23_Co) phases, which was more conducive to promoting the nucleation and refinement of the Al_13_Fe_4_ phase. Moreover, the compromised stability of Al_78_(Fe_23_Cr) and Al_78_(Fe_23_Mn) phases promoted the formation of other metastable phases, e.g., Al_6_Fe, in the Al-Fe alloy after the substitution of Cr and Mn.

(3) The comparison of Al-5wt%Fe-1wt%Cr and Al-5wt%Fe-1wt%Ni alloys revealed that the addition of Ni significantly refined the Al_13_Fe_4_ phase, whereas Cr mainly improved the morphology of the Al_13_Fe_4_ phase without refining the grains of the Al_13_Fe_4_ phase.

These preliminary results demonstrate that the selection of an optimal alloying element to promote nucleation and refinement, is of great significance to further understand the grain refinement mechanism and provide theoretical bases for the application of *M*-doped Al-Fe alloys.

## Figures and Tables

**Figure 1 materials-14-00768-f001:**
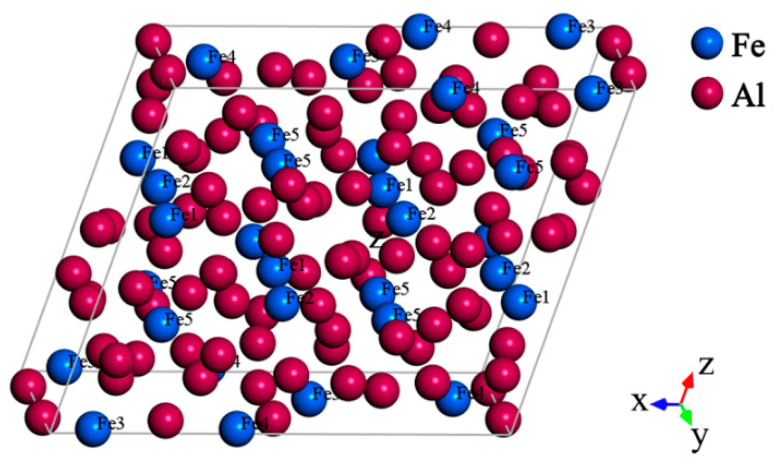
The unit cell of the Al_13_Fe_4_ compound.

**Figure 2 materials-14-00768-f002:**
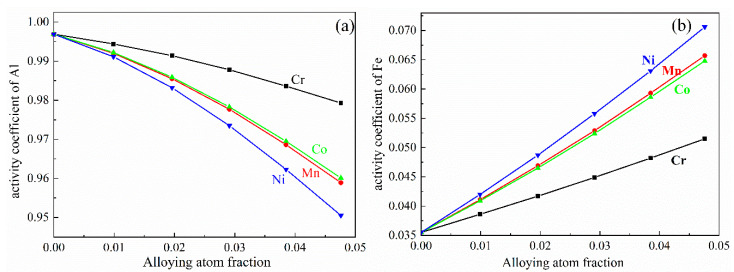
The activity coefficient of (**a**) Al and (**b**) Fe atoms in Al-2.47 at% Fe alloy melt at 850 °C.

**Figure 3 materials-14-00768-f003:**
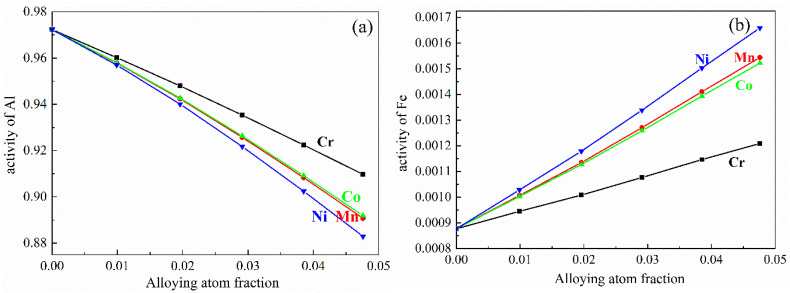
The activity of (**a**) Al and (**b**) Fe atoms in Al-2.47 at% Fe alloy melt at 850 °C.

**Figure 4 materials-14-00768-f004:**
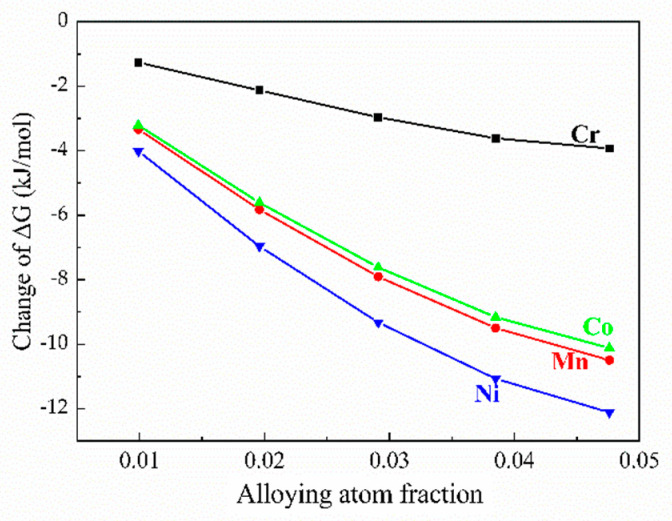
Influence of M content on the change in Gibbs free energy of Al_13_Fe_4_ phase.

**Figure 5 materials-14-00768-f005:**
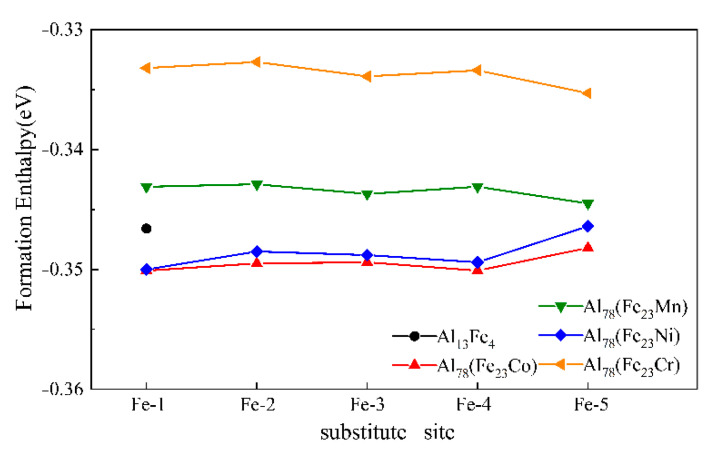
The formation enthalpy of Al_13_Fe_4_ and Al_78_(Fe_23_M) phases, where M = Cr, Mn, Co and Ni.

**Figure 6 materials-14-00768-f006:**
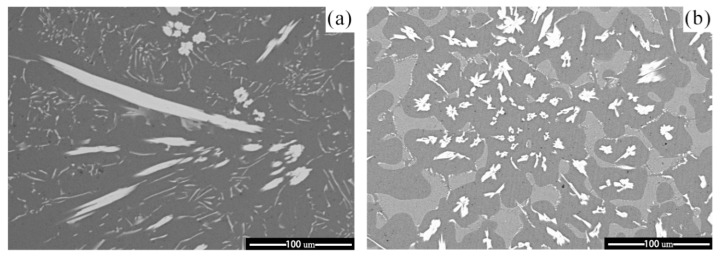
The as-cast microstructure of (**a**) Al-5wt%Fe, (**b**) Al-5wt%Fe-1wt%Ni and (**c**) Al-5wt%Fe-1wt%Cr.

**Figure 7 materials-14-00768-f007:**
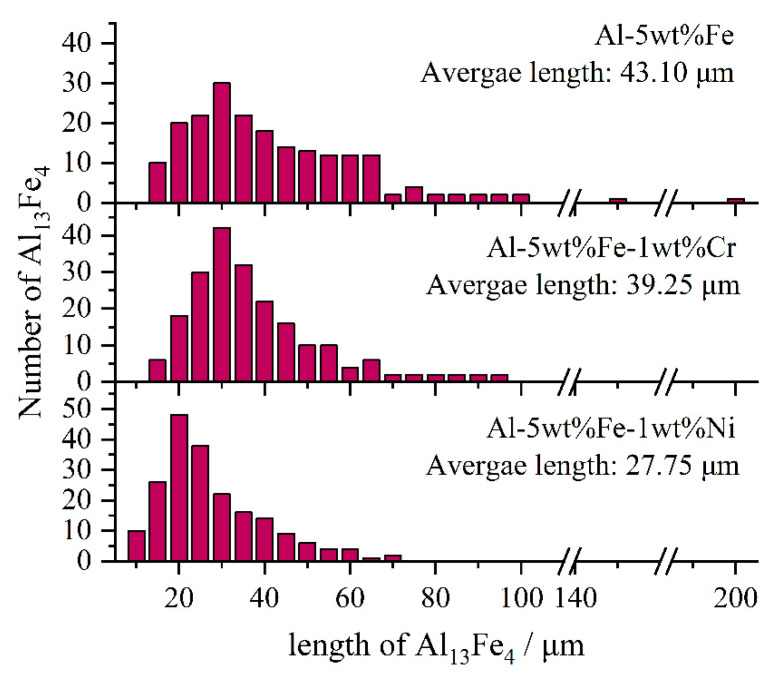
Lengths distribution of Al_13_Fe_4_ in Al-5wt%Fe-M alloys.

**Figure 8 materials-14-00768-f008:**
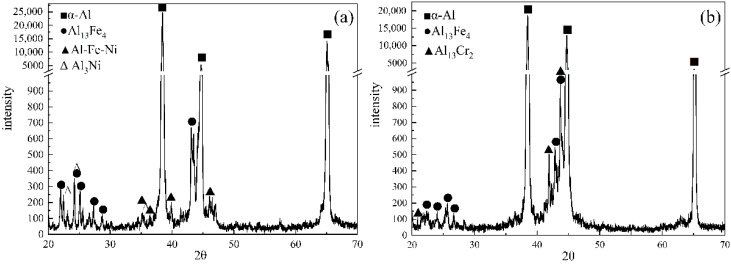
XRD patterns of as-cast alloys: (**a**) Al– 5wt%Fe–1wt%Ni and (**b**) Al-5wt%Fe-1wt%Cr.

**Figure 9 materials-14-00768-f009:**
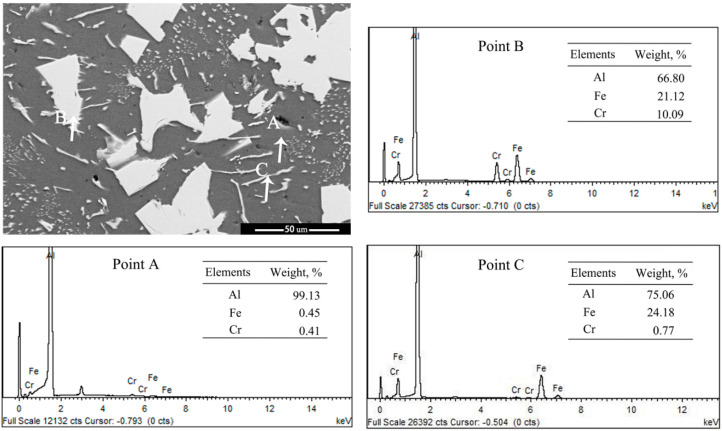
SEM image and corresponding EDS spectra of Point−A, −B and −C of Al-5wt%Fe-wt%Cr alloy.

**Figure 10 materials-14-00768-f010:**
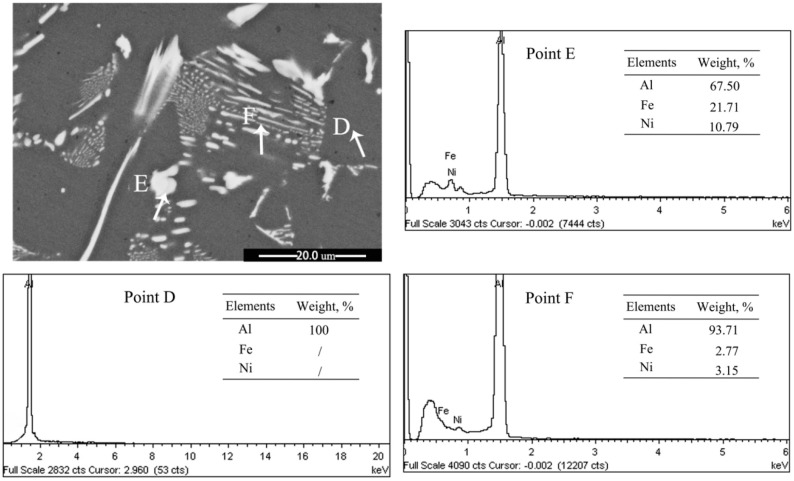
SEM image and corresponding EDS spectra of Point−D, −E and −F of Al–5wt%Fe-1wt%Ni alloy.

**Table 1 materials-14-00768-t001:** Atomic parameters in A1_13_Fe_4_ [22].

Atom	x/a	y/b	z/c	No. in Cell	Atom	x/a	y/b	z/c	No. in Cell
Al1	0	0.5	0.5	2	Al11	0.1883	0.2164	0.1111	8
Al2	0	0.2441	0	4	Al12	0.3734	0.2110	0.1071	8
Al3	0.3223	0	0.2778	4	Al13	0.1765	0.2168	0.3343	8
Al4	0.2352	0	0.5392	4	Al14	0.4959	0.2832	0.3296	8
Al5	0.0812	0	0.5824	4	Al15	0.3664	0.2238	0.4799	8
Al6	0.2317	0	0.9729	4	Fe1	0.0865	0	0.3831	4
Al7	0.4803	0	0.8277	4	Fe2	0.4018	0	0.6243	4
Al8	0.3100	0	0.7695	4	Fe3	0.0907	0	0.9890	4
Al9	0.0869	0	0.7812	4	Fe4	0.4001	0	0.9857	4
Al10	0.0645	0	0.1730	4	Fe5	0.3188	0.2850	0.2770	8

**Table 2 materials-14-00768-t002:** The calculation parameters of different alloying elements [31].

Element	$n_{ws}^{1/3}$ ((d.u.))^1/3^	*φ* (V)	*V*^2/3^ (cm^2^)	*u*	*r/p*	*T_m_* (°C)
Al	1.39	4.20	4.64	0.07	1.9	660 [32]
Fe	1.77	4.93	3.69	0.04	1.0	1536 [33]
Cr	1.73	4.65	3.74	0.04	1.0	1875 [34]
Mn	1.61	4.45	3.78	0.04	1.0	1252 [35]
Co	1.75	5.10	3.50	0.04	1.0	1495 [34]
Ni	1.75	5.20	3.50	0.04	1.0	1455 [36]

**Table 3 materials-14-00768-t003:** The calculated values of Wilson’s parameters in binary M-Al, M-Fe, and Al-Fe systems at 850 °C.

M (alloy)	A_M/Al_	A_Al/M_	A_M/Fe_	A_Fe/M_	A_Al/Fe_	A_Fe/Al_
Cr	−3.1812	−3.7626	−0.5424	−0.5546	−4.1184	−3.5930
Mn	−6.0446	−7.1179	0.0850	0.0840		
Co	−5.9119	−7.0274	−0.2059	−0.2147		
Ni	−7.0740	−8.3139	−0.5676	−0.5872		

**Table 4 materials-14-00768-t004:** The lattice constants of Al_13_Fe_4_ and Al_78_(Fe_23_M) phases, where M = Cr, Mn, Co, and Ni.

Compound	*a* (Å)	*b* (Å)	*c* (Å)	*V_cell_* (Å^3^)
Al_13_Fe_4_	15.366	8.012	12.393	1453.9
Al_78_(Fe_23_Ni)	15.367	8.027	12.385	1455.3
Al_78_(Fe_23_Co)	15.355	8.020	12.392	1454.1
Al_78_(Fe_23_Mn)	15.393	8.005	12.396	1455.5
Al_78_(Fe_23_Cr)	15.383	8.023	12.413	1459.5

## Data Availability

Data is contained within the article.

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
