# Peer review of "Influence of Cr, Mn, Co and Ni Addition on Crystallization Behavior of Al13Fe4 Phase in Al-5Fe Alloys Based on ThermoDynamic Calculations"

_materials, 2021, doi:10.3390/ma14040768_

Round 1

Reviewer 1 Report

Dear Authors

The article is interesting. I did not notice any factual errors. There are technical errors that can be corrected.
1) No citation of item 44 .
2) Wrong citation order (line 89) - correct.

Reviewer 2 Report

I suggest to put in order the citation cause you cited “Fan et al. [18-20] have” then you have number [17]

Please provide details of preparation of “The metallographic samples” cause there is suggested only electropolishing, but this is the last step so in order to replicate it is necessary all the steps

Please provide details for SEM and EDS measurements because there is nothing about it

“is shown in Figure 2 and Figure 3.” Say OK but how I can understand this activity ??? please provide details…apart you suggested in table 1 and present table 2 and table 2 but this were in different section and difficult to follow. I suggest discussing each table and Figure separately and introduce after the text it follow

The results reveal that the incorporation of alloying elements reduced the Gibbs free energy and, in turn, increased the driving force for phase transition. May you are right but for a nontechnical person it is difficult to see that the Gibbs free energy is reduced as you don’t provide any number to validate this, also for the increased driving force…please be more explicit

 You discuss Figure 6 by using SEM images but there is no clear details of grain, normally they can be clearly detected with EBSD or backscattered imaging not simple SEM.

Otherwise you have to point out with some arrow which tip of morphology you have detected

A section of discussion is necessary cause there is no discussion about your results

Reviewer 3 Report

In their manuscript Pang and coauthors report results of a joint experimental-theoretical investigation of the properties of Al-5Fe alloys. In particular, employing a combination of experimental techniques and computer simulations they studied the nfluence of Cr, Mn, Co and Ni addition on the crystallization behavior.

The paper is well-written and well-organized. The reported results are obtained using standard experimental techniques and interpreted within the standard density functional theory approach demonstrating a very good agreement between theory and experiment.

The manuscript leaves a good overall impression. I only have a few minor points:

In Table 1 melting temperatures should be supported by proper references. Melting temperature of Co and Cr (Phys. Rev. B 63, 132104 (2001)), Fe (J. Phys.: Conf. Ser. 121 022018 (2008)), Al (Earth and Planetary Science Letters 153, 3–4, 223-227 (1997)), Mn (Journal of Applied Physics 108, 033517 (2010)). Ni (Phys. Rev. B 87, 054108 (2013)).

Give details on crystal structure, space group and unit-cell parameters.

The crystal structure has “15 different Al atoms and 5 different Fe atoms” it means they occupy different atomic positions since all Al are identical. The same for Fe atoms. It would be better to add the complete crystal structure (a Table) including atomic positions. Otherwise is difficult to know what are Fe1 … Fe5.

The cut-off energy was set at 400 eV. Is this enough? Did you run a convergence test?

Did you optimize only unit-cell parameters or also atomic positions? Please comment on it.

Please compare computed unit-cell parameters with experimental values.

Cu kα1 target was monochromatic?

Please us larger fonts in figures.

Give unit-cell parameters in Angstrom not in nm.

Experimental values should always have errors.

Provide the energy of SEM.

Round 2

Reviewer 2 Report

Thank you

This manuscript is a resubmission of an earlier submission. The following is a list of the peer review reports and author responses from that submission.

Round 1

Reviewer 1 Report

The authors have studied the influence of Cr, Mn, Co and Ni on the activity of Fe and Al, Gibbs free energy of Al13Fe4 and Formation enthalpy of Al13Fe4. The main objective behind this study was to understand the effect of the above alloying elements on the crystallisation behaviour of Al13Fe4 phase in Al-5Fe alloys. The authors have used a combination of experimental and theoretical methods for their investigations.

Overall, the investigations and main conclusions seem reasonable. However, I would like to understand few things more clearly before taking any decision. My main questions/comments for the authors are as follows:

(1) The authors found that different alloying elements, i.e., Cr, Mn, Co and Ni influence the activity of Fe and Al differently. Unfortunately, I couldn't find an explanation behind this behaviour. Can the authors provide an explanation why different elements have different influence on the activity? Even a qualitative explanation will help me and the readers to understand the observation more clearly.

(2) In page 6, the authors mentioned that coordination number influence site occupation of Cr, Mn, Co and Ni. While I am not doubting the finding, I don't understand the reason behind this finding. Can the authors explain how coordination number is influencing the site occupation? What is the underlying reason?

(3) Out of the four elements Cr, Mn are antiferromagnetic in nature while Co and Ni are ferromagnetic. What kind of magnetic states did the authors consider for the alloying elements? How did the choice of magnetic state of the alloying elements influence the formation enthalpy?

(4) The authors used first-principles calculations to determine formation enthalpy but used an empirical model to estimate Gibbs free energy. In principle, first-principles calculations can provide reliable estimation of Gibbs free energy. Why didn't the authors use this approach for the calculation of Gibbs free energy?

(5) A related question to the previous comment. The model used by the authors employs some physical parameters for the estimation of Gibbs free energy. How does this model address entropic contributions due to finite temperature excitations such as vibrational, magnetic etc? I have difficulty to understand the role of temperature dependent entropic contributions in the model adopted by the authors.

(6) The authors must provide more details on experimental measurements. For instance, they must include the details of microstructure measurements.

Reviewer 2 Report

The influence of M-element addition on the activity of Al and Fe atoms is investigated. Although the topic is interesting, the paper lacks in details and looks in an initial state. The reviewer believes that the actual paper quality is not sufficient for publication in such a high-reputed journal. For this reason, it is suggested to reject the paper. Authors are encouraged in performing a deep improvement, a better correlation of experimental and theoretical results, a complete reorganization, presentation of the results and resubmit the paper as a new submission.

The main issues are:

The paper is unclear and confusing. Moreover the English form should be improved, it is hard to follow the manuscript as it is presented.

The performed experimental tests are poorly duscussed, how many tests were performed? Was a statistical analysis carried out?

The mechanical properties of the obtained alloys were not investigated.

A deep discussion of the comparison  between theretical and experimental model should be made. Was a validation of the theoretical models performed? If yes, how? It is not clear within the paper.

Minor issues:

Many parameters are not introduced within the text.

Nomenclature must be added.

Line 83: place reference after Fan et al

Tm is a temperature measured in kelvin, “K” should be used instead of “k”

Use only one temperature unit (K or °C) in the whole manuscript

Terms used in the equations are poorly mentioned and references are used for their description. However a little  explanation of terms and discussion would help the reader in a better understanding of the content.